

# Feeding habits of four-finger threadfin fish, *Eleutheronema tetradactylum,* and its diet interaction with co-existing fish species in the coastal waters of Thailand

Teuku Haris Iqbal[1], Sukree Hajisamae[1], Apiradee Lim[1], Sitthisak Jantarat[1], Wen-Xiong Wang[2] and Karl W.K. Tsim[3]

[1] Faculty of Science and Technology, Prince of Songkla University, Pattani, Thailand
[2] City University of Hong Kong, Kao Loon, Hong Kong
[3] Hong Kong University of Science and Technology, Kaoloon, Hong Kong

Corresponding author
Sukree Hajisamae, sukree.h@psu.ac.th

## ABSTRACT

This study assessed the feeding habits of four-finger threadfin fish, *Eleutheronema tetradactylum*, and its diet relationship with other fish species in the tropics. Fish samples were collected from four locations along the coastal regions of Thailand. A whole year field sampling event was conducted to investigate the diet relationship of threadfin fish with other ten co-existing fish species in Pattani Bay during January 2021 and January 2022. *E. tetradactylum* was an active and specific predator with significant diet shift during ontogeny. Specifically, the juvenile fish fed largely on zooplankton especially *Acetes*/shrimp postlarvae, and small sized-fish fed on penaeid shrimps, while medium and large-sized fish shifted their diets to a combination of penaeid shrimp, fish and squid. Size and sex of fish as well as site of collection significantly affected gut fullness index and average number of food type ($p < 0.05$). Transitional sex fish predated almost entirely on other fishes (87.2%), whereas male and female fish fed mainly on penaeid shrimp (66.5%) and other fish (51.3%), respectively. Fish size and mouth opening controlled the size of prey, with the larger fish with larger mouth-opening fed primarily on the larger size of prey. Moreover, *E. tetradactylum* shared its diets inclusively with *Epinephelus coioides, Johnius belangerii, Scomberomorus commerson, Scomberoides lysan, Otolithes ruber* and *Lutjanus russelli*. Penaeid shrimp and teleost fish were the main food types shared by these fishes. This study provided important information on the feeding habits of *E. tetradactylum* and its diet relationship with other co-existing fish species living in the same habitat of a tropical coastal region.

## INTRODUCTION

Feeding ecological study in fishes has been conducted worldwide due to its fundamental importance in understanding the role of fish in ecosystem (*e.g., Wootton, 1990; Blaber, 2013; Baeta & Ramon, 2013; Kwak, Klumpp & Park, 2015; Ende et al., 2018; Lin et al., 2020; Dinh et al., 2021; Soe et al., 2021; Hajisamae et al., 2022). Eleutheronema tetradactylum,*

known as four-finger threadfin, belongs to the family Polynemidae and is mostly found in the estuarine and marine shallow water habitat (*Motomura et al., 2002*; *Nesarul et al., 2014*; *Adelir-Alves et al., 2018*; *Roshni et al., 2021*). It distributes along the tropical and subtropical coasts of Indo-West Pacific region, from the Gulf of Persia to the Northern part of Australia (*Motomura et al., 2002*; *Ballagh et al., 2012*; *Hena et al., 2011*; *Adelir-Alves et al., 2018*). Adult of this species is a typical-carnivore feeding on crustacean and fish (*Hena et al., 2011*). They consume mainly copepods, mysids, and crustacean larvae during larval stage (*Malhotra, 1953*), but shift to shrimp and teleost fish when they become juvenile (*Haywood et al., 1998*; *Jaferian et al., 2010*; *Soe et al., 2021*). However, most previous reports did not focus on *E. tetradactylum* and only examined a particular stage of fish based on the availability of collected samples. Moreover, size, sex and maturity stage of fish as well as habitat and season are among the most important factors affecting the feeding characteristics of fish (*Hajisamae, Chou & Ibrahim, 2004*; *Pereira et al., 2016*; *Nanami, 2018*; *Chuaykaur et al., 2020*). None of these earlier studies assessed directly on *E. tetradactylum*. The threadfin fish is hermaphrodite with sex change occurring during growth, *i.e.,* early-stage is male and turns to female when it reaches the size of 40 cm at about 2 years (*Pember, 2006*; *Shihab et al., 2017*); it is therefore necessary to assess the feeding habits of fish during sex-switching process. Moreover, most fish species undergo dietary shift along their ontogeny but the timing of change varies between fish species (*Blaber, 2000*). This goal of such change is to maximize the energy intake, enhance the growth rate and minimize the risk of predation (*Brown, 1985*), and is normally associated with season, habitat and morphological characteristics (*Labropoulou & Eleftheriou, 1997*). The dietary ontogenetic changes of *E. tetradactylum* have not been reported. Therefore, understanding the change of this species will fulfill the gap of current scientific knowledge.

Habitat and season are the essential factors structuring trophic organization and feeding habits of fishes (*Hajisamae, Chou & Ibrahim, 2004*; *Soe et al., 2021*). Some fishes may choose a habitat providing abundant and diverse prey while some fish may select a habitat with less prey abundance but easy to capture (*Crowder & Cooper, 1982*). To gain a better understanding of the feeding ecology of fish, it is essential to assess the inter-specific trophic relationships of fishes that co-exist in the same habitat (*Elliott et al., 2007*; *Soe et al., 2021*). Several fishes compete for the same food type and some avoid competing with other co-existing species (*Vogt et al., 2017*; *Páez-Rosas et al., 2018*; *Soe et al., 2021*). However, some fishes may predate on other or similar fish species as prey. This piscivores behavior is common among fish community (*Gerking, 1994*). Therefore, investigating the pattern of relationship between fish community in similar habitat is important to understand the ecological function of each fish species.

The coastal regions of Southeast Asia are known for their high biodiversity and abundance of resources including a variety of fish species (*e.g., Hajisamae, Chou & Ibrahim, 2004*; *Hajisamae & Yeesin, 2014*; *Fazrul et al., 2020*; *Soe et al., 2021*). However, few studies emphasized on fish diet and trophic relationship between fish community (*Hajisamae & Ibrahim, 2008*; *Hajisamae, 2009*; *Hajisamae, Fazrul & Pradit, 2015*; *Islam et al., 2018*; *Paul et al., 2018*). In Thailand, threadfin fishes are captured by local fishermen from both the Gulf of Thailand and the Andaman Sea. The mouth of Pattani Bay is locally known as one

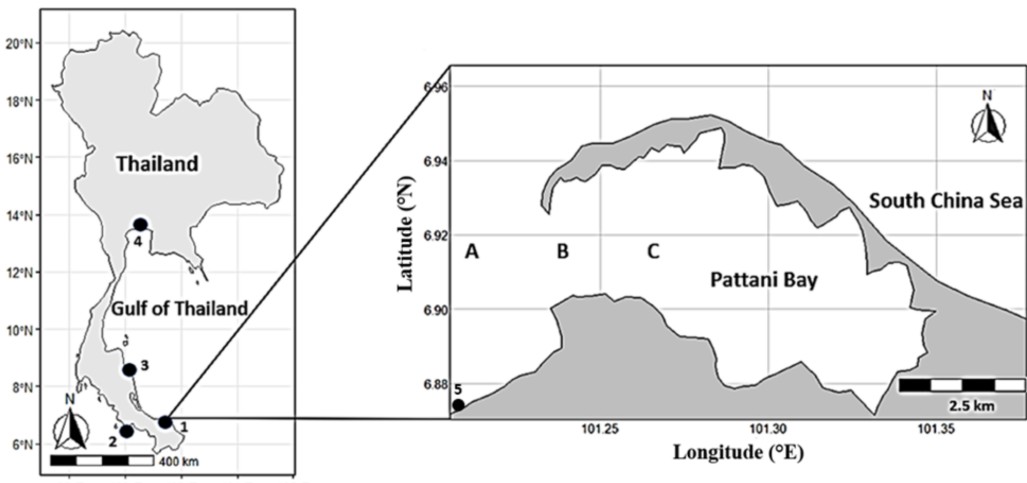

**Figure 1** **Map of the study area along coastal waters of Thailand (left) and Pattani bay (right).** 1, Pattani; 2, Satun; 3, Nakhon Sri Thammarat; 4, Samut Prakan; A, B and C, sampling sites in Pattani bay.

of the important fishing grounds for *E. tetradactylum* in Thailand. Traditional fishermen normally employ the threadfin fish gill nets to capture this fish as their main source of income. Thus, this habitat is appropriate as a specific study site for this species. The aims of this study were to assess the feeding habits and diet relationship of *E. tetradactylum* and its association with other fish species in coastal habitat.

## MATERIAL AND METHODS

### Pattani Bay and fish sampling

The major sampling in this study was conducted in Pattani Bay, Thailand. The bay covers approximately 74 km², located in the lower part of the Gulf of Thailand at Latitude 06°52′5.3″N and Longitude 101°15′0.3″E (Fig. 1). Based on the rainfall, the bay has three seasons; dry season from January to May, moderate rainfall season (southwest monsoon) from May to September, and the heavy rainy season (monsoon) from September to December. Three sampling sites (A, B and C) along the bay mouth area, locally known as threadfish fish fishing fleets, were selected.

Monthly fish sampling was conducted for 13 months from January 2021 to January 2022 at three study sites with 2 km apart in Pattani Bay. Three sets of traditional monofilament gill nets as replicates (mesh size of 4.5 cm stretched, 5 m deep, and 540-meter total length) were simultaneously employed at each site by three traditional fishing boats and left floating for 60 min before being hauled onboard. To cover all three sites, the sampling was conducted between 0500-0900. Samples of *E. tetradactylum* were removed from the nets, all died after attached to the nets, and preserved with ice. Ten other fish species caught by the nets together with *E. tetradactylum*, including *Scomberomorus commerson* (narrow barred Spanish mackerel), *Eubleekeria jonesi* (Jones' pony fish), *Hilsa kelee* (kelee shad), *Johnius belangerii* (belanger's croaker), *Lutjanus russellii* (Russel's snapper), *Otolithes ruber* (tiger tooth croaker), *Parastomateus niger* (black pomfret), *Scatophagus argus* (spotted scat),

*Scomberoides lysan* (doublespotted queenfish) and *Epinephelus coioides* (orange-spotted grouper), were also removed and prepared by the same methods. They were brought back to the Faculty of Science and Technology, Prince of Songkla University, sorted and identified immediately. Up to 30 individual fish per species were randomly collected from each of the three sampling sites, and preserved with 10% formalin for four days before being transferred to 70% ethanol for further analyses. This study was approved by institutional animal care and use committee, Prince of Songkla University, Thailand, and the submitted protocol met the criteria of the Exempt Determination Research (MHESI 68014/605 Ref. Ex.001/2020).

## Fish sampling from other areas

Additional threadfin fish samples were directly collected from fisherman in the provinces of Satun, Nakorn Si Thammarat and Samut Prakan provinces to cover all coastal regions of Thailand. A total of up to 25 fish samples, where possible, from each province were preserved with ice and brought back to the laboratory, where they were immediately preserved with 10% formalin for four days before being transferred to 70% ethanol for further analyses (*Hajisamae, 2009*).

## Fish/prey dimension and diet analysis

A total of 541 samples of *E. tetradactylum* and 251 specimens of other fish species collected from the Pattani Bay were used. Additional 140 samples of *E. tetradactylum* from other areas including the provinces of Satun, Nakorn Si Thammarat and Samut Prakan were included in the analysis. They were initially measured for standard length (cm) and separated into 4 size classes: juvenile fish (<13.0 cm), small-sized fish (13.1–24.0 cm), medium-sized fish (24.1–35.0 cm) and large-sized fish (>35.1 cm). Sexes were separated based on the appearances of testis for male, ovarian for female and transitional gonad for transitional sex. Average standard lengths for male, female and transitional sex fishes used for this study were $19.04 \pm 8.13$ cm, $29.75 \pm 5.04$ and $29.21 \pm 2.28$ cm, respectively. Additionally, vertical mouth opening ($M_V$) and horizontal mouth opening ($M_H$) of *E. tetradactylum* were measured. Mouth opening area (MA) of fish was calculated based on *Erzini et al. (1997)* and *Kyritsi & Moutopoulos (2018)*.

$$MA = 0.25\pi (M_V \times M_H).$$

Where MA = mouth opening area, $M_V$ = vertical mouth opening, $M_H$ = horizontal mouth opening and $\pi$ is 3.14.

The surgical scissor was applied to remove the entire gut. The gut fullness index value (FL) was assessed on a scale of 0 (empty stomach) to 6 (fully distended with food) that was expressed as empty, trace, $\frac{1}{4}$ full, $\frac{1}{2}$ full, $\frac{3}{4}$ full, full, and gorged stomachs (*Pillay, 1952*; *Hajisamae, 2009*; *Soe et al., 2021*). As most of the fish samples were freshly caught from nature before preservation, the stomach contents were still fresh with low level (<30%) of physical digestion. They were later identified to the lowest possible taxon. To quantify the diet compositions of *E. tetradactylum*, a modified point method and a numerical method were applied. For the modified point method, each food component was given points

(%) proportionated to its estimation to volume in the scaled petri discs or plates (*Hyslop, 1980*; *Hajisamae, 2009*; *Soe et al., 2021*). Larger diets were visually assessed based on the relative contribution on the scaled petri dish or plate. Small items and zooplankton were assessed under a stereo microscope. Complete food components were also simultaneously identified and counted for numerical analysis. A relative volumetric contribution (%V), relative numerical contribution of each food type (%N) and relative frequency of occurrence (%FO) were calculated. For all fish species collected from Pattani Bay, only a modified point method was applied and later converted to volumetric contribution (%V) for further analysis. Very small items such as phytoplankton component and detritus were clarified and assessed under a compound light microscope.

Three selected major diets including fish, shrimp and squid found in the stomachs of *E. tetradactylum* were measured for length (cm) and body height (cm). Only completely intact or remaining prey were included in measurement to assess the predator–prey relationship (*Kwak, Klumpp & Park, 2015*). Each specific prey had different methods of calculating the dimension. For fish, the length was based on standard length, and the height was the greatest body depth between dorsal and ventral side. Shrimp was measured by straightening out and the length calculated from the rostrum to the edge uropod part (tail fan), and the height was the depth between carapace. The length of squid was measured from the edge of the head part to posterior end of the mantle, and the height was between the largest depth of mantle. The prey dimensions were later calculated by multiplying length and the height of prey (*Karpouzi & Stergiou, 2003*; *Paul et al., 2017*).

## Data analysis

Several diet attributes including average gut fullness index (FL), average number of food type (AF), vacuity index (VI), and diet breadth (*Bi*) were calculated to assess the contribution of each food type in the diets of different fish species (*Hyslop, 1980*; *Hajisamae, 2009*). The diet attributes were also calculated for each factor including fish size class, sex, study site and season for *E. tetradactylum*. Average gut fullness index (FL) is an average of the relative gut fullness of all guts examined. Dominant food type is the type of food found in the greatest proportion, based on %V data, in the guts. Vacuity index (VI) is the number of empty guts as a percentage of the total number of guts. The Index of Relative Importance (IRI) is calculated based on *Pinkas (1971)*, *Hyslop (1980)* and *Osman, El Ganainy & Amin (2019)* and applied only for *E. tetradactylum*.

$$\mathrm{IRI}_V = (\%N + \%V) \times \%FO$$

where $\mathrm{IRI}_V$ = Index of Relative Importance by volumetric, %N = percentage of numerical, %V = percentage of volume and %FO = percentage frequency of occurrence. Percentage of Index of Relative Importance (%IRI$_i$) is applied for *E. tetradactylum* and expressed as the percentage for each food group based on *Cortés (1997)* and *Osman, El Ganainy & Amin (2019)*.

$$\%\mathrm{IRI}_i = \frac{\mathrm{IRI}_i}{\sum_{i=1}^{n} \mathrm{IRI}_i} \times 100$$

where %$IRI_i$ = percentage of Index of Relative Importance, $i$ = number of specific food category and $n$ = the total number of food categories.

Diet breadth ($Bi$) is assessed, based on %V data, by the formula of Levin's standardized index (*Krebs, 1989*; *Labropoulou & Papadopoulou-Smith, 1999*).

$$Bi = \left(\frac{1}{n-1}\right)\left(\left(\frac{1}{\sum_{i,j=1}^{n} P_{ij}^2}\right) - 1\right)$$

where $B_i$ = the Levin's standard index for predator '$i$', '$P_{ij}$' = the proportion of diet of predator '$i$' that is made up of food '$j$' and '$n$' = the number of food category.

Diet overlap ($C_H$) is calculated to assess dietary inter-specific relationship between different fish species inhabiting Pattani Bay. Simplified Morisita-Horn index (*Horn, 1966*), based on %V, data is calculated for diet overlap between fish species:

$$C_H = \frac{2\left(\sum p_{ij} p_{ik}\right)}{\sum p_{ij}^2 + \sum p_{ik}^2}$$

where '$C_H$' = Morisita-Horn index of overlap between species '$j$' and '$k$'; '$p_{ij}$' = the proportion of food '$i$' of the total food quantity by species '$j$'; '$p_{ik}$' = the proportion of food '$i$' of the total food used by species '$k$'. The degree of overlap was classified as low overlap (0.0–0.29), moderate overlap (0.30–0.59) and high overlap (0.60–1.00) (*Langton, 1982*).

Trophic level (TL) is calculated, based on %V, to assess the trophic rank of fishes based on the proportion of each prey component in fish diet using the following equation:

$$TL = 1 + \sum_{i=1}^{n}(V_i * T_i)$$

where $TL$ = trophic level, $V_i$ = the proportion of prey "$i$" in the diet of fish and $T_i$ = the mean $TL$ of prey based on *Froese & Pauly (2000)*.

## Statistical analysis

One-way analysis of variance (ANOVA) was used to analyze whether size class, sex, season and study site affected the fullness index (FL) and average number of food type (AF) in the stomach *E. tetradactylum*. Once the difference is detected, Tukey's range test was applied. The raw data were Log (X+1) transformed to reduce the non-normality prior to statistical tests. Regression analyses were conducted to analyze the relationship between fish standard length (SL in cm) with mouth opening area (MA) of *E. tetradactylum*, standard length with overall prey dimension and specific prey dimension, mouth area with overall prey dimension and specific prey dimension (*Erzini et al., 1997*; *Paul et al., 2017*; *Kyritsi & Moutopoulos, 2018*). The raw data were square root transformed prior to analysis. Cluster analysis based on Bray-Curtis similarity matrix (%) and group average linkage method was conducted to assess how *E. tetradactylum* and other co-existing species interacted with each other for food. A cluster dendogram was created among eleven fish species caught together with *E. tetradactylum* in Pattani Bay. Analysis of similarity (ANOSIM) was later used to assess the significant level of the difference between groups. Additionally, a similarity

percentage (SIMPER) was used to identify type of food responsible for the formation of each cluster group on the dendogram (*Hajisamae & Yeesin, 2014*; *Soe et al., 2021*). All these statistical tests were performed by PRIMER software package version 5.0 based on %V data (*Clarke & Gorley, 2001*).

## RESULTS

### Diet compositions and trophic attributes of *E. tetradactylum*

In general, *E. tetradactylum* predated mainly on penaeid shrimp, (%IRI = 50.20%), fish (33.28%) and *Acetes* sp./shrimp post larvae (8.41%). Details of food compositions based on %N, %FO, %V and %IRI of *E. tetradactyum* from all sites along the coast of Thailand are shown in Table 1. The %IRI of penaeid shrimp were 76.81%, 84.70% and 70.02% in the stomachs of fish from Samut Prakan, Satun, and Nakorn Si Thammarat Provinces, respectively (Table 2). About 70% of the penaeid shrimp consumed was *Fenneropenaeus merguiensis*. Samples of *E. tetradactylum* collected from Pattani Bay fed mainly on a combination of fish (%V = 43.79%), followed by penaeid shrimp, especially *F. merguiensis* (35.68%) (Table 3). The prey fish were mainly Clupeiformes, Perciformes, Mugilliformes and Atheriniformes. These included *Ambassis sp.*, *Atherinomorus lacunosus*, *Dussumieria elopsoides*, *Encrasicholina devisi*, *Hilsa kelee*, *Leiognathus equulus*, *Planiliza subviridis*, *Stolephorus commersonii*, *Thryssa setirostris* and *Trichiurus lepturus*. Different food compositions between size class, sex, season and site of collection were observed. Details of food compositions and attributes for all size class, sex, season and study site are illustrated in Tables 2 and 4. The main food types of different sex stages were observed. The male and female fishes fed mainly on shrimp and fish, respectively, but fish at transitional sex stage predated almost entirely on other fishes. Results of analysis of variance (ANOVA) indicated that value of fullness index (FL) varied significantly based on size class, sex, season and study site (Table 5). Average number of food type (AF) was influenced by fish size, sex and study site but not by season (Table 5). Results of post-hoc analysis are given in Table 6. The FL value of transitional sex fish was not different from male or female fishes, but the variation of AF between transitional sex fish ($1.09 \pm 0.30$) and female ($1.53 \pm 0.65$) was significant ($p = 0.006$).

### Ontogenetic shift of diet compositions

Based on samples from Pattani Bay, juvenile fish fed largely on *Acetes*/shrimp post larvae (%IRI = 69.54%), followed by fishes (21.89%) and polychaete (4.78%) (Table 2). The small sized-fish turned its diet to penaeid shrimp (71.30%) with some proportions of fish (21.61%) and squid (1.18%). In the medium-sized fish, the reduction of penaeid shrimp was observed to 43.62% and the increment of fish and squid was found to be 43.97% and 4.40%, respectively. Importance of penaeid shrimp reduced gradually when the fish grew to larger size (24.20%), with a concurrent increase of fish and squid (30.12% and 30.34%, respectively).

**Table 1  Diet compositions (%IRI) of *P. tetradactylum* collected from four study sites along the coasts of Thailand (*n* = 541).**

| Food type | Diet compositions (%) | | | |
|---|---|---|---|---|
| | % *N* | %FO | %V | %IRI |
| Shrimp and *Acetes* | | | | **59.10** |
| *Acetes* spp./shrimp post larvae | 13.51 | 13.55 | 7.85 | 8.89 |
| Penaeid shrimp | 21.59 | 22.91 | 31.86 | 50.20 |
| Fishes | | | | **33.28** |
| *Ambassis* sp. | 2.52 | 2.31 | 2.63 | 0.49 |
| *Atherinomorus lacunosus* | 0.37 | 0.58 | 0.73 | 0.03 |
| *Dussumieria elopsoides* | 0.28 | 0.29 | 0.22 | 0.01 |
| *Eleutheronema tetradactylum* | 0.37 | 0.58 | 0.59 | 0.02 |
| *Encrasicholina devisi* | 2.62 | 2.59 | 2.92 | 0.59 |
| *Hilsa kelee* | 0.19 | 0.14 | 0.10 | 0.00 |
| *Leiognathus equulus* | 9.72 | 3.89 | 4.14 | 2.21 |
| *Planiliza subviridis* | 0.37 | 0.29 | 0.37 | 0.01 |
| *Stolephorus commersonnii* | 2.15 | 2.74 | 2.95 | 0.57 |
| *Thryssa setirostris* | 0.37 | 0.29 | 0.28 | 0.01 |
| *Trichiurus lepturus* | 0.75 | 0.58 | 0.74 | 0.04 |
| Fish bone | 14.86 | 14.55 | 11.73 | 15.87 |
| Fish flesh | 10.93 | 13.40 | 13.55 | 13.45 |
| Mantis shrimp | | | | **0.01** |
| *Odontodactylus scyllarus* | 0.09 | 0.14 | 0.19 | 0.01 |
| Mollusca | | | | **2.78** |
| *Corbula fortisulcata* | 0.84 | 1.30 | 0.72 | 0.08 |
| *Uroteuthis* spp. | 4.49 | 5.76 | 6.95 | 2.70 |
| Polychaete | | | | **0.18** |
| *Perinereis nuntia* | 1.40 | 1.44 | 1.71 | 0.18 |
| UDF | 5.70 | 7.20 | 7.30 | 3.84 |
| Zooplankton | 2.43 | 2.45 | 1.83 | 0.43 |
| Plastic debris | 2.43 | 3.03 | 0.64 | 0.38 |

**Notes.**
[a]Bold value indicates the sum of each diet group.

## Relationship between standard length and mouth area of *E. tetradactylum* with prey dimensions

A total of 128 fish samples were analyzed with the standard length of 5.90 cm and maximum length of 46.11 cm. The maximum and minimum mouth areas of fish sample were 32.20 cm$^2$ and 0.26 cm$^2$, respectively. The slope of regression showed a significantly positive relationship between standard length and mouth area ($R^2 = 0.88$, $P < 0.001$), indicating that mouth opening of the fish was positively related to the standard length (Fig. 2A).

Both standard length and mouth area showed positive relationships with overall prey dimensions ($R^2 = 0.3281$, $P < 0.001$; $R^2 = 0.3594$, $P < 0.001$), indicating that the larger fish consumed the larger prey (Figs. 2B and 2C). Results of a specific analysis on the relationship between standard length and mouth opening with prey dimensions of three

Iqbal et al. (2023), *PeerJ*, DOI 10.7717/peerj.14688

Peerj

**Table 2  Food composition of *E. tetradactylum* (%IRI) of different, size classes, sex, season and site collected from all sites along the coasts of Thailand (*n* = 541).** Season factor was only for Pattani Bay (*n* = 347).

| Food | Size | | | | Sex | | | Season | | | Site | | | | |
|---|---|---|---|---|---|---|---|---|---|---|---|---|---|---|---|
| | J | S | M | L | M | F | T | D | M | R | PB | BA | SP | ST | NA |
| Samples (*n*) | 90 | 199 | 233 | 19 | 402 | 128 | 11 | 83 | 171 | 93 | 347 | 24 | 16 | 12 | 142 |
| Shrimp and *Acetes* | **69.54** | **74.94** | **45.85** | **24.20** | **70.49** | **41.83** | **6.38** | **66.48** | **47.22** | **64.07** | **49.03** | **14.62** | **76.81** | **84.70** | **75.36** |
| *Acetes* spp./ shrimp post larvae | 69.54 | 3.63 | 2.22 | 0.00 | 4.01 | 2.45 | 3.67 | 1.56 | 15.31 | 6.89 | 13.35 | 0.00 | 0.00 | 0.00 | 5.35 |
| Penaeid shrimp | 0.00 | 71.30 | 43.62 | 24.20 | 66.48 | 39.38 | 2.71 | 64.91 | 31.91 | 57.18 | 35.68 | 14.62 | 76.81 | 84.70 | 70.02 |
| Fishes | **21.89** | **21.61** | **43.97** | **30.12** | **22.69** | **51.33** | **87.25** | **25.48** | **42.85** | **29.53** | **43.78** | **15.70** | **22.27** | **14.16** | **19.91** |
| *Ambassis* sp. | 0.00 | 0.05 | 1.78 | 0.00 | 0.31 | 0.93 | 0.00 | 7.94 | 0.07 | 0.08 | 1.22 | 0.00 | 0.00 | 0.00 | 0.00 |
| *A. lacunosus* | 0.00 | 0.04 | 0.03 | 0.00 | 0.02 | 0.02 | 0.00 | 0.06 | 0.01 | 0.03 | 0.04 | 0.00 | 0.00 | 0.00 | 0.02 |
| *D. elopsoides* | 0.00 | 0.00 | 0.03 | 0.00 | 0.01 | 0.00 | 0.00 | 0.22 | 0.00 | 0.00 | 0.01 | 0.00 | 0.00 | 0.00 | 0.00 |
| *E. tetradactylum* | 0.00 | 0.01 | 0.02 | 0.83 | 0.00 | 0.17 | 0.00 | 0.00 | 0.14 | 0.00 | 0.00 | 0.00 | 13.26 | 0.00 | 0.00 |
| *E. devisi* | 0.00 | 0.37 | 1.30 | 0.00 | 0.47 | 0.72 | 0.00 | 0.93 | 0.22 | 0.83 | 0.58 | 0.00 | 2.63 | 0.00 | 0.61 |
| *H. kelee* | 0.00 | 0.00 | 0.01 | 0.00 | 0.00 | 0.02 | 0.00 | 0.00 | 0.01 | 0.00 | 0.00 | 0.00 | 1.00 | 0.00 | 0.00 |
| *L. equulus* | 0.00 | 0.44 | 6.26 | 1.03 | 0.28 | 13.72 | 2.71 | 0.18 | 5.91 | 0.79 | 5.11 | 0.00 | 1.13 | 0.00 | 0.00 |
| *P. subviridis* | 0.00 | 0.00 | 0.04 | 0.00 | 0.00 | 0.02 | 0.00 | 0.00 | 0.05 | 0.00 | 0.00 | 0.00 | 0.00 | 1.83 | 0.00 |
| *S. commersonnii* | 0.00 | 0.24 | 1.30 | 0.61 | 0.32 | 1.36 | 0.00 | 0.45 | 0.44 | 0.67 | 0.77 | 0.00 | 0.00 | 0.00 | 0.51 |
| *T. setirostris* | 0.00 | 0.00 | 0.04 | 0.00 | 0.01 | 0.00 | 0.00 | 0.29 | 0.00 | 0.00 | 0.02 | 0.00 | 0.00 | 0.00 | 0.00 |
| *T. lepturus* | 0.00 | 0.02 | 0.08 | 0.00 | 0.06 | 0.00 | 0.00 | 1.31 | 0.00 | 0.00 | 0.09 | 0.00 | 0.00 | 0.00 | 0.00 |
| Fish bone | 0.85 | 11.42 | 24.50 | 8.71 | 10.13 | 22.26 | 62.53 | 9.17 | 17.04 | 15.75 | 17.46 | 11.39 | 3.29 | 12.33 | 10.88 |
| Fish digest | 21.04 | 9.01 | 8.58 | 18.93 | 11.07 | 12.10 | 22.01 | 4.93 | 18.95 | 11.38 | 18.49 | 4.31 | 0.95 | 0.00 | 7.90 |
| Mantis shrimp | **0.00** | **0.00** | **0.01** | **0.00** | **0.00** | **0.02** | **0.00** | **0.00** | **0.01** | **0.00** | **0.00** | **0.00** | **0.00** | **0.00** | **0.00** |
| *O. scyllarus* | 0.00 | 0.00 | 0.01 | 0.00 | 0.00 | 0.02 | 0.00 | 0.00 | 0.01 | 0.00 | 0.00 | 0.00 | 0.00 | 0.00 | 0.00 |
| Mollusks | **0.00** | **1.19** | **4.63** | **30.67** | **2.46** | **2.66** | **2.71** | **2.88** | **2.70** | **2.61** | **2.41** | **30.50** | **0.00** | **1.14** | **1.32** |
| *C. fortisulcata* | 0.00 | 0.01 | 0.23 | 0.34 | 0.02 | 0.24 | 2.71 | 0.58 | 0.06 | 0.02 | 0.08 | 1.36 | 0.00 | 0.00 | 0.02 |
| *Uroteuthis* spp. | 0.00 | 1.18 | 4.40 | 30.34 | 2.44 | 2.42 | 0.00 | 2.30 | 2.65 | 2.59 | 2.34 | 29.14 | 0.00 | 1.14 | 1.30 |
| Polychaete | **4.78** | **0.00** | **0.00** | **0.00** | **0.30** | **0.00** | **0.00** | **0.00** | **1.14** | **0.00** | **0.46** | **0.00** | **0.00** | **0.00** | **0.00** |
| *P. nuntia* | 4.78 | 0.00 | 0.00 | 0.00 | 0.30 | 0.00 | 0.00 | 0.00 | 1.14 | 0.00 | 0.46 | 0.00 | 0.00 | 0.00 | 0.00 |
| UDF | 2.04 | 1.44 | 4.86 | 15.01 | 3.20 | 3.51 | 3.67 | 4.03 | 5.90 | 2.00 | 3.58 | 39.17 | 0.92 | 0.00 | 1.61 |
| Zooplankton | 1.68 | 0.62 | 0.03 | 0.00 | 0.59 | 0.04 | 0.00 | 0.00 | 0.14 | 1.17 | 0.28 | 0.00 | 0.00 | 0.00 | 1.30 |
| Plastic debris | 0.07 | 0.21 | 0.66 | 0.00 | 0.27 | 0.61 | 0.00 | 1.13 | 0.04 | 0.62 | 0.44 | 0.00 | 0.00 | 0.00 | 0.49 |

**Notes.**

Size: J, juvenile; S, small; M, medium; L, large.
Sex: M, male; F, female; T, transitional sex.
Season: D, dry; M, moderate rainy; R, rainy.
Site: PB, Pattani bay; BA, Bangtawa; SP, Samut Prakan province; ST, Satun province; NA, Nakorn Si Tammarat province.
UDF, Unidentified food.
[a]Bold value indicates the sum of each diet group.

**Table 3  Diet compositions (%IRI) of *E. tetradactylum* and co-existing species in Pattani Bay collected by gill net from January 2021 to January 2022.**

| Species/Feed | # sample | Food types | | | | | | | | | | | |
|---|---|---|---|---|---|---|---|---|---|---|---|---|---|
| | | shr | sqd | pol | zoo | phy | det | alg | biv | crb | fis | udf | dig |
| E. tetradactylum | 347 | **36.25** | 6.23 | 2.66 | 1.28 | 0.00 | 0.00 | 0.00 | 0.51 | 0.00 | **30.51** | 6.83 | 15.15 |
| S. commerson | 15 | **32.00** | 0.00 | 0.00 | 4.33 | 0.33 | 0.00 | 0.00 | 0.00 | 0.00 | **45.00** | 0.00 | 18.33 |
| E. jonesi | 15 | 0.00 | 0.00 | 4.69 | 15.31 | **29.38** | 11.25 | 0.00 | 11.88 | 0.00 | **22.50** | 0.00 | 5.00 |
| H. kelee | 109 | 7.14 | 0.00 | 0.70 | 11.12 | 6.30 | **57.14** | 0.00 | 0.00 | 1.39 | **12.00** | 0.00 | 4.22 |
| J. belangerii | 19 | **31.05** | 0.00 | 0.00 | **25.53** | 2.63 | 0.00 | 0.00 | 0.00 | 16.58 | 16.05 | 0.00 | 8.16 |
| L. russellii | 21 | **46.43** | 0.00 | 4.05 | 0.00 | 0.00 | 0.00 | 0.00 | 0.00 | 15.95 | **24.05** | 0.00 | 9.52 |
| O. ruber | 21 | **30.24** | 0.00 | 0.00 | 0.00 | 0.00 | 9.52 | 0.00 | 3.10 | 2.38 | **53.81** | 0.00 | 0.95 |
| P. niger | 12 | **16.67** | 0.00 | 2.50 | **48.75** | 10.42 | 0.00 | 0.00 | 6.25 | 0.00 | 15.42 | 0.00 | 0.00 |
| S. argus | 13 | 0.00 | 0.00 | 0.00 | 21.54 | 11.54 | **37.69** | 17.69 | 0.00 | 0.00 | 11.54 | 0.00 | 0.00 |
| S. lysan | 13 | 14.62 | **16.54** | 0.00 | 11.54 | 0.00 | 0.00 | 0.00 | 0.00 | 0.00 | **36.54** | 0.00 | 20.77 |
| E. coioides | 13 | **28.08** | 0.00 | 0.00 | 1.15 | 0.38 | 0.00 | 0.00 | 1.54 | 14.62 | **42.31** | 0.00 | 11.92 |

Notes.

[a]Bold value indicate greater contribution on diet composition.

*n*, number of samples; shr, shrimp; sqd, squid; pol, polychaeta; zoo, zooplankton; phy, phytoplankton; det, detritus; alg, algae; biv, bivalve; crb, crab; fis, fishes; udf, unidentified food; dig, digested item.

**Table 4  Diet attributes of *E. tetradactylum* and ten co-existing fish species in Pattani Bay collected by gill net from January 2021 to January 2022.**

| Species | Average size of fish (SL and TL in cm ± SD) | Average of fullness index (FL ± SD) | Vacuity index (VI) | Total food type (TL) | Average number of food type (AF ± SD) | Diet breadth (*Bi*) | Trophic level (*TL*) |
|---|---|---|---|---|---|---|---|
| E. tetradactylum | 21.78 ± 8.70 | 3.22 ± 1.75 | 32.72 | 8 | 1.35 ± 0.57 | 0.41 | 3.67 |
| S. commerson | 32.25 ± 1.65 | 3.27 ± 0.88 | 34.78 | 5 | 1.80 ± 0.94 | 0.48 | 3.89 |
| E. jonesi | 11.38 ± 1.15 | 2.50 ± 0.89 | 20.00 | 7 | 1.81 ± 0.75 | 0.70 | 2.37 |
| H. kelee | 18.68 ± 1.70 | 3.08 ± 1.23 | 21.01 | 7 | 1.54 ± 0.67 | 0.25 | 1.31 |
| J. belangerii | 20.81 ± 1.80 | 2.53 ± 1.39 | 17.39 | 6 | 1.74 ± 0.45 | 0.70 | 2.86 |
| L. russellii | 21.85 ± 3.03 | 2.86 ± 1.15 | 27.59 | 5 | 1.43 ± 0.60 | 0.56 | 3.34 |
| O. ruber | 24.73 ± 5.94 | 2.95 ± 0.92 | 25.00 | 6 | 1.71 ± 0.46 | 0.31 | 4.25 |
| P. niger | 15.85 ± 2.55 | 3.08 ± 1.00 | 25.00 | 6 | 2.00 ± 0.60 | 0.46 | 2.31 |
| S. argus | 10.33 ± 1.01 | 2.92 ± 1.04 | 18.75 | 5 | 1.92 ± 0.76 | 0.76 | 1.77 |
| S. lysan | 30.90 ± 2.66 | 2.92 ± 0.64 | 18.75 | 5 | 1.85 ± 0.55 | 0.80 | 3.82 |
| E. coioides | 24.62 ± 1.59 | 3.08 ± 1.12 | 7.14 | 7 | 2.38 ± 0.51 | 0.40 | 4.14 |

major prey types (fish, squid and shrimp) are shown in Figs. 3, 4 and 5. The dimensions of the consumed prey items increased significantly for all prey types with the increase of standard length and mouth opening of the predator. This confirms that the mouth dimension of this species controls the size of all three prey types (fish, squid and shrimp).

## Inter-specific diet relationship between *E. tetradactylum* and other co-existing species

A total of 792 fish samples from eleven fish species were examined (Table 3). Three species feeding on shrimp as the main food item included *E. tetradactylum, J. belangerii* and *L.*

**Table 5** Results of analysis of variance (ANOVA) on the fullness index (FL) and average number of food type (AF) ($n = 541$) in relations to fish size, sex, season and site. Seasonal factor was only used for Pattani Bay collected by gill net from January 2021 to January 2022 ($n = 347$).

| Diet attributes | Factors | df | MS | F | P-value |
|---|---|---|---|---|---|
| Fullness index (FL) | Size | 3 | 0.938 | 13.82 | **<0.001** |
| | Sex | 2 | 0.282 | 3.924 | **0.020** |
| | Season | 2 | 0.483 | 6.796 | **0.001** |
| | Site | 4 | 0.450 | 6.437 | **<0.001** |
| Average number of food type (AF) | Size | 3 | 0.377 | 19.99 | **<0.001** |
| | Sex | 2 | 0.416 | 21.45 | **<0.001** |
| | Season | 2 | 0.008 | 0.351 | 0.704 |
| | Site | 4 | 0.029 | 5.896 | **<0.001** |

Notes.
[a]Bold value indicates significant difference.

**Table 6** Results of Tukey's post-hoc analysis of the fullness index and average number of food type ($n = 541$) in relations to fish size, sex, season and site factors. Season factor was only applied for Pattani bay ($n = 347$).

| | Fullness index | | | | Average number of food type | | | |
|---|---|---|---|---|---|---|---|---|
| **Size** | **Juvenile** | **Small** | **Medium** | | **Juvenile** | **Small** | **Medium** | |
| Small | 0.971 | | | | 0.761 | | | |
| Medium | **<0.001** | **<0.001** | | | **<0.001** | **<0.001** | | |
| Large | **0.003** | **0.002** | 0.4235 | | **<0.001** | **<0.001** | 0.385 | |
| Sex | Male | Female | | | Male | Female | | |
| Fe | **0.007** | | | | **<0.001** | | | |
| Tr | 0.449 | 0180 | | | 0.436 | **0.006** | | |
| **Season** | **Dry** | **Moderate rainy** | | | **Dry** | **Moderate rainy** | | |
| Moderate rainy | 0.275 | | | | 0.853 | | | |
| Rainy | **<0.001** | **0.007** | | | 0.361 | 0.380 | | |
| **Site** | **PB** | **BA** | **SP** | **ST** | **PB** | **BA** | **SP** | **ST** |
| PB | | | | | | | | |
| BA | 0.305 | | | | 0.350 | | | |
| SP | 0.371 | 0.242 | | | 0.872 | 0.888 | | |
| ST | 0.953 | 0.788 | 0.965 | | 0.346 | 0.444 | 0.937 | |
| NA | **<0.001** | **0.003** | **0.014** | 0.296 | **<0.001** | 0.152 | 0.936 | 0.432 |

Notes.
Site: PB, Pattani bay; BA, Bangtawa; SP, Samut Prakan province; ST, Satun province; NA, Nakorn Si Thammarat province.
[a]Bold value indicates significant difference.

*russellii*. Fish was considered a dominant prey in the stomachs of *S. commerson*, *O. ruber*, *S. lysan* and *E. coioides*. Two species including *H. kelee* and *S. argus* fed dominantly on detritus. Zooplankton and phytoplankton were the dominant food for *P. niger* and *E. jonesi*, respectively.

Results of trophic attributes of eleven fish species including FL, VI, total food type, average number of food type and B$_i$ are provided in Table 4. The FL ranged from 2.50 ± 0.89 for *E. jonesi* to 3.27 ± 0.88 for *S. commerson*, VI varied from 7.14 for *E. coioides* to 34.78 for *S. commerson*, total number of food type was between 5 for the three fish species

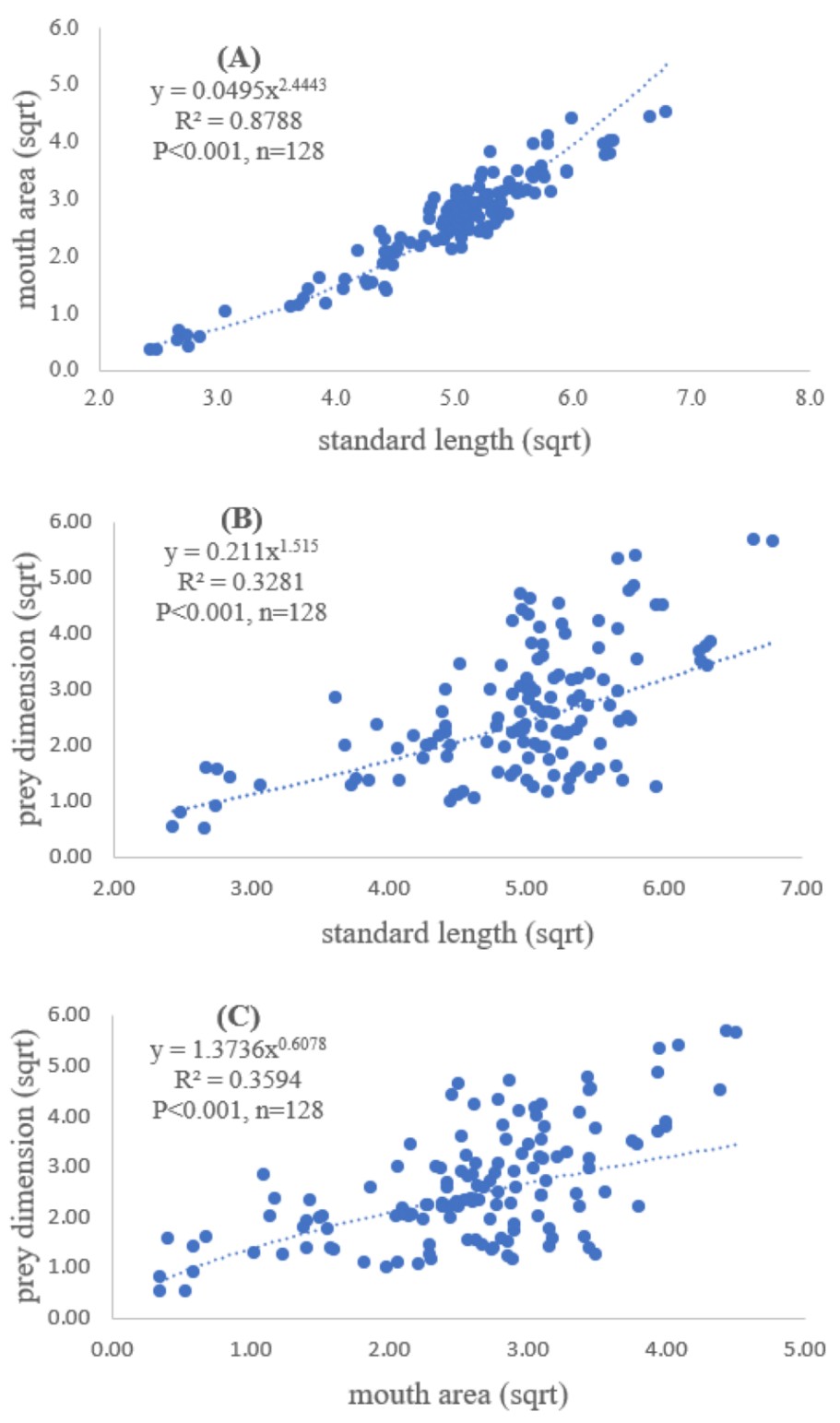

**Figure 2** Relationship between (A) standard length (cm) of *E. tetradactylum* and mouth area (cm²), (B) standard length and overall prey dimension (cm²) and (C) mouth area and overall prey dimension.

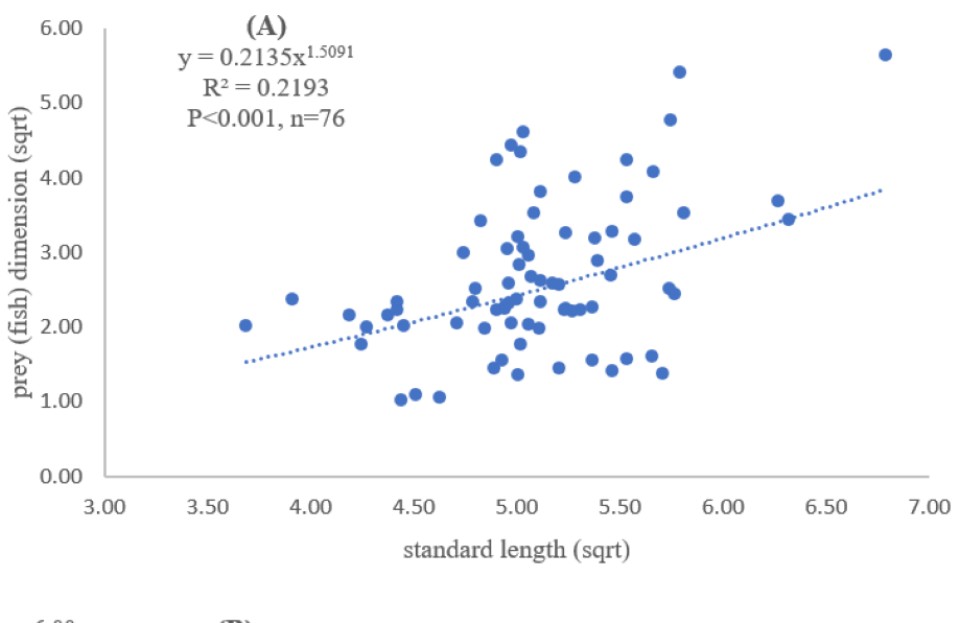

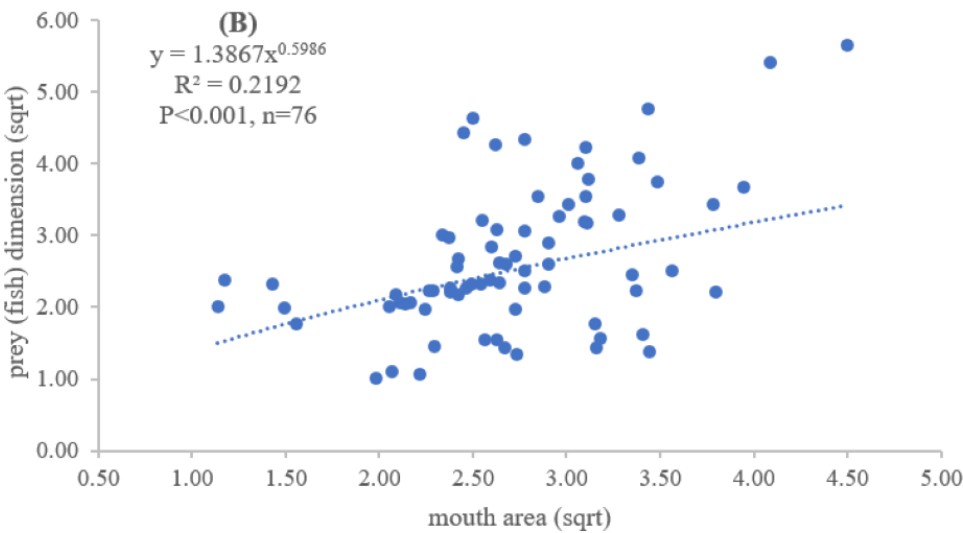

**Figure 3** Relationship between (A) standard length (cm) of *E. tetradactylum* and prey (fish) dimension (cm²) and (B) mouth area (cm²) and prey (fish) dimension (cm²).

to 8 for *E. tetradactylum*, AF was from 1.35 ± 0.57 for *E. tetradactylum* to 2.38 ± 0.51 for *E, coioides*, and $B_i$ ranged from 0.25 for *H. kelee* to 0.80 for *P. lysan*. The TL values ranged from 1.31 for *H. kelee* to 4.25 for *O. ruber*.

Based on the data from Pattani Bay, twenty-three out of fifty-five pairs of analyses were shown to have high diet overlaps ($C_H$ >0.60) (Table 7). The $C_H$ values for *E. tetradactylum* indicated significant overlaps with six co-existing species including *S. commerson, J. belangerii, L. russellii, O. ruber, S. lysan* and *E. coioides*. The low $C_H$ values, considering no overlap for food, were found between *E. tetradactylum* and *E. jonesi, H. kelee, P. niger* and *S. argus*.

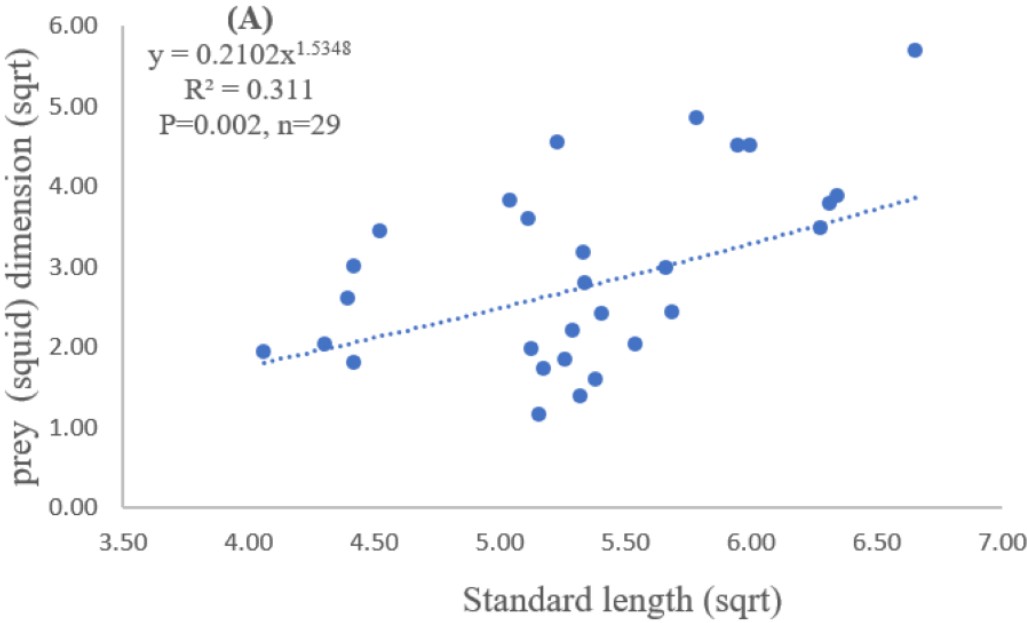

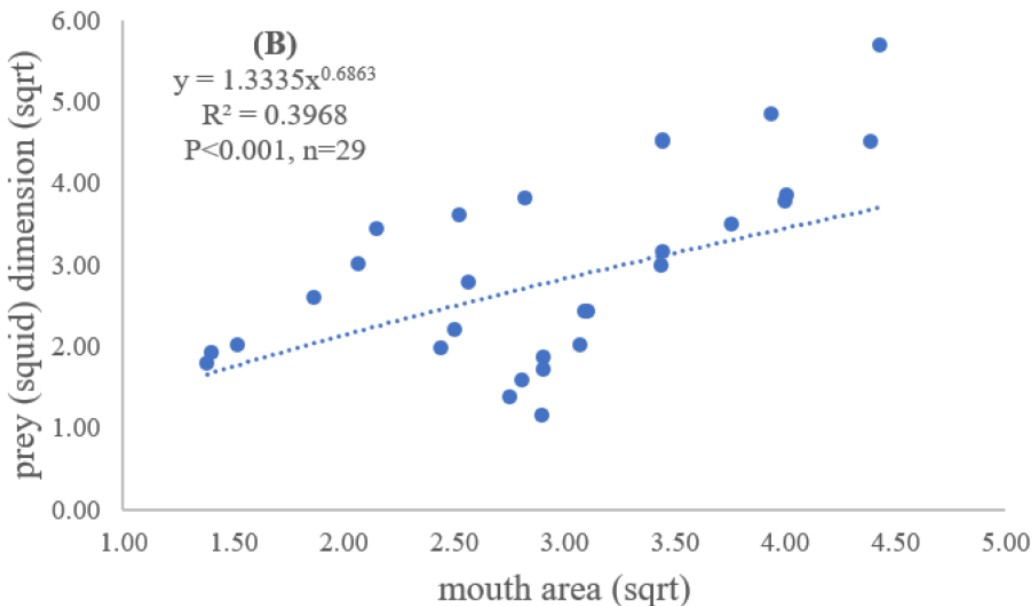

**Figure 4** Relationship between (A) standard length (cm) of *E. tetradactylum* and prey (squid) dimension (cm²) and (B) mouth area (cm²) and prey (squid) dimension (cm²).

The above result was also confirmed by the cluster analysis. Two trophic clusters, at 45% similarity, were formed by eleven fish species caught together with *E. tetradactylum* (Fig. 6). Result of analysis of similarity (ANOSIM) indicated the difference of diet composition among the two groups ($P = 0.003$, Global $R = 0.907$). The first group (G1) included *O. ruber, L. russellii, E. coioides, J. belangerii, S. commerson, S. lysan* and *E. tetradactylum*. A similarity percentage (SIMPER) indicated that teleost fish (47.8% contribution) and

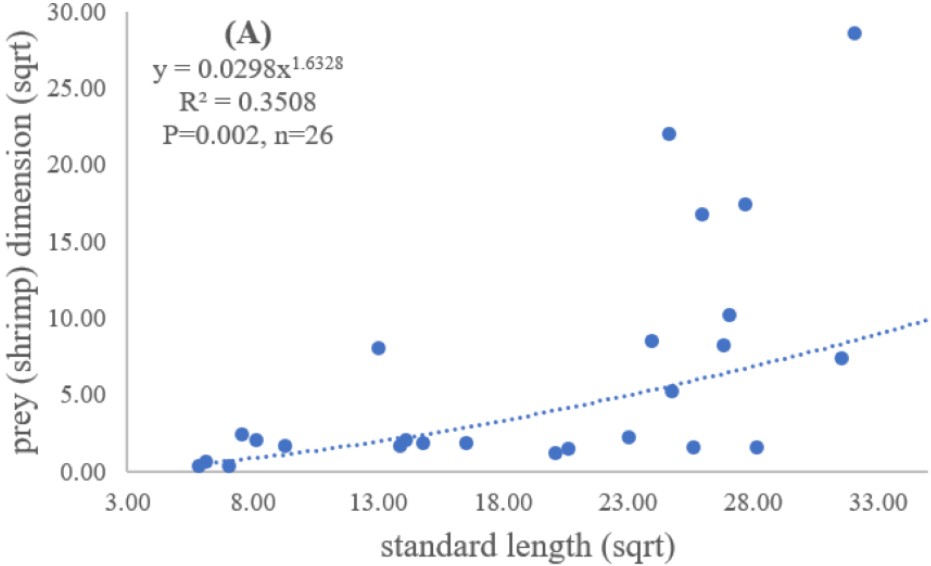

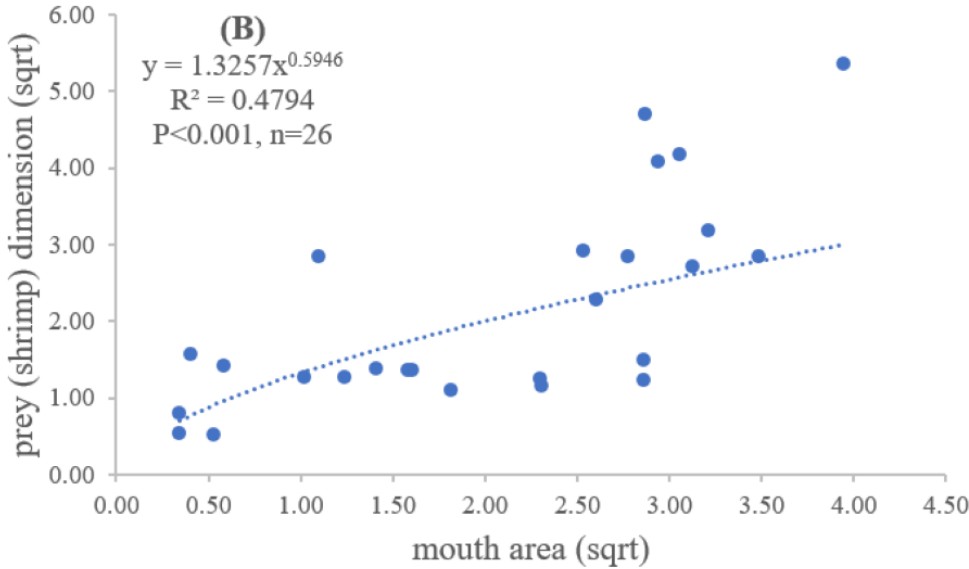

**Figure 5** Relationship between (A) standard length (cm) of *E. tetradactylum* and prey (shrimp) dimension (cm²) and (B) mouth area (cm²) and prey (shrimp) dimension (cm²).

shrimp (44.5%) were the main food types contributing to the formation of this group. The second cluster (G2) was formed by four fish species including *P. niger, E. jonesi, S. argus* and *H. kelee*. Contribution of zooplankton, teleost fish, detritus and phytoplankton to the formation of this group were 29.6%, 25.7%, 21.0% and 17.89%, respectively.

**Table 7** Diet overlap of *E. tetradactylum* and 10 co-existing fish species collected by gill net in Pattani Bay from January 2021 to January 2022.

| Species | 1 | 2 | 3 | 4 | 5 | 6 | 7 | 8 | 9 | 10 |
|---|---|---|---|---|---|---|---|---|---|---|
| *E. tetradactylum* (1) | - | | | | | | | | | |
| *S. commerson* (2) | **0.94** | | | | | | | | | |
| *E. jonesi* (3) | 0.36 | 0.44 | | | | | | | | |
| *H. kelee* (4) | 0.23 | 0.37 | 0.46 | | | | | | | |
| *J. belangerii* (5) | **0.74** | **0.70** | 0.42 | 0.26 | | | | | | |
| *L. russellii* (6) | **0.91** | **0.84** | 0.24 | 0.20 | **0.82** | | | | | |
| *O. ruber* (7) | **0.85** | **0.93** | 0.47 | 0.37 | 0.60 | **0.78** | | | | |
| *P. niger* (8) | 0.41 | 0.45 | 0.60 | 0.27 | **0.77** | 0.38 | 0.39 | | | |
| *S. argus* (9) | 0.15 | 0.21 | **0.62** | **0.85** | 0.33 | 0.10 | 0.31 | 0.49 | | |
| *S. lysan* (10) | **0.84** | **0.88** | 0.51 | 0.25 | **0.65** | **0.64** | **0.77** | 0.50 | 0.10 | |
| *E. coioides* (11) | **0.90** | **0.95** | 0.44 | 0.24 | **0.74** | **0.88** | **0.93** | 0.40 | 0.19 | **0.83** |

**Notes.**
[a] Bold value indicates high overlap.

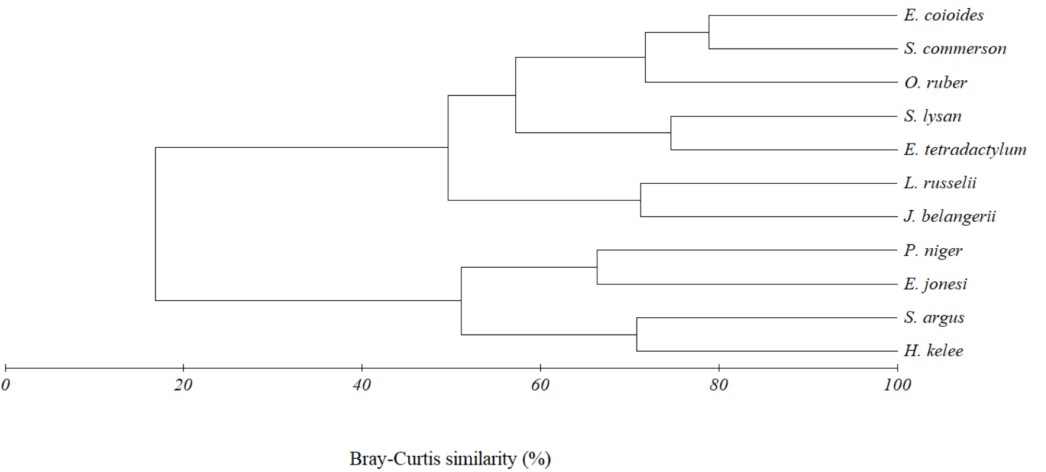

**Figure 6** Cluster dendrogram indicating trophic group of eleven fish species caught together with *E. tetradactylum* in Pattani Bay during January 2011–January 2022.

## DISCUSSION

This study, considered as the first report, identified the comprehensive feeding ecology of *E. tetradactylum* from the juvenile stage to the large-sized adult stage along the whole transexual change. It confirmed that *E. tetradactylum* is an active and specific predator. The juvenile fed mainly on zooplankton especially *Acetes* and shrimp post larvae, small sized-fish predated mainly on penaeid shrimp, whereas medium and large-sized fishes predated more on a combination of penaeid shrimp, fish and squid. The predation on fish and penaeid shrimp as the main food items of the adult was reported earlier (*Salini, Brewer & Blaber, 1998*; *Farmer & Wilson, 2011*). In the Northwest of Australia, the fish fed mainly on teleost fish (79%) and crustacean (18%) (*Farmer & Wilson, 2011*). In the Norman-river estuary,

Gulf of Carpentaria, Australia, fishes of sizes 14.3–53.1 cm fed mainly on fish (82.7%) and penaeid shrimp (14.8%) (*Salini, Brewer & Blaber, 1998*). Results from the present study indicated that fish foods were mainly Clupeiformes, Perciformes, Mugilliformes and Atheriniformes. The occurrence of cannibalism is observed in this study as *E. tetradactylum* was also found in the stomach of this species. It is noted that the main proportion of penaeid shrimp predated by this fish was *F. merguensis*. This shrimp species distributed abundantly and was easy to be captured in the area where the threadfish fish inhabited, or the fish particularly searched for this shrimp to predate. Based on the practice of local fishermen, the fishing ground for threadfin fish is normally in the same fishing fleet with *F. merguensis*.

Ontogenetic change during ontogeny, an important strategy for the survival of a species, was observed for *E. tetradactylum* in this study (*Baldo & Drake, 2002*; *Braga, Bornatowski & Vitule, 2012*; *Monteiro, Contente & Faria, 2018*; *Chuaykaur et al., 2020*). Diet may change due to variations in habitat, competition and predation risk (*Gerking, 1994*). The timing of the change is usually when the juveniles become sub-adults or adults and leave the nursery habitats, which is normally in the nearshore areas, to the deeper waters. Juvenile stage of this species fed initially on *Acetes*/shrimp post larvae and other zooplankton. However, the proportion of *Acetes*/shrimp post larvae reduced significantly when the fish grew larger to the small-sized class. The role of penaeid shrimp with some combination of fish as food become more obvious at this size class. Later, penaeid shrimp prey reduced slightly in the medium-sized class and fishes became the important food with some combination of squid. For the larger-sized fish, apart from penaeid shrimp and fish as the major prey types, squid was found significantly in the stomach.

Even though feeding on a wide range of zooplankton during juvenile stage is common for many marine and brackish water fish species (*Gning, Vidy & Thiaw, 2008*; *Hajisamae, Yeesin & Ibrahim, 2006*; *Hajisamae, 2009*; *Chuaykaur et al., 2020*), a very high proportion of *Acetes*/shrimp postlarvae as food of juvenile is a particular diet niche of *E. tetradactylum*. The foraging for tiny crustaceans by juvenile of this species was reported in two different regions. *Nasir (2000)* found that juvenile fish (6.5–9.2cm) in the inshore waters of Khor Al-Zubair, Northwest Persian Gulf fed completely on small shrimp. Another study by *Premcharoen (2014)* indicated that juvenile and small sized-fishes inhabiting mangrove area in the inner Gulf of Thailand fed mainly on sergestid shrimp/*Acetes* (40.1%), shrimp (24.3%) and zooplankton (20.9%). This indicated a specific diet niche for juvenile of *E. tetradactylum*. The small-sized fish predated mainly on shrimp with some combinations of fish and small portion of zooplankton as a continuous feeding mode from juvenile stage which shifted from *Acetes*/shrimp postlarvae to the larger shrimp. *Soe et al. (2021)* reported that the small sized-stage of this species (18.7 ± 22.2 cm) fed mainly on shrimp (70.2%) and teleost fish (26.1%). The medium and large-sized fishes then gradually reduced shrimp as food and predated more on fish and squid. Additionally, it is known that this fish is a protandrous hermaphrodite with sex change during growth from male to female, *i.e.*, early-stage is mature as male and becomes female when they reach the size of about 40 cm at age of about 2 years (*Pember, 2006*; *Shihab et al., 2017*). The present study recorded food during its sex transient period, during which they fed almost entirely on other fishes as their food (87.2%), whereas male and female fishes fed on penaeid shrimp (66.5%) and

other fish (51.3%), respectively, as their main food. With a lower value in the transitional period, it appeared that fish of this stage was more dietary specific compared to the female fish. Moreover, the size of fish for the transitional period was bigger than the male but was almost the same size as the female examined. Therefore, size of fish affected the change of food during ontogeny for for male or female instead of those during transitional period. The reason and mechanism responsible for this strong piscivorous behavior during this period required for further investigation.

Both predator and prey size determined the success of foraging and fish diet (*Persson, 1990*; *Scharf, Juanes & Rountree, 2000*). The relationship between fish size, mouth opening, and prey dimensions for *E. tetradactylum* was first examined in this study similar to other fish species (*i.e.*, *Erzini et al., 1997*; *Paul et al., 2017*), the mouth opening of *E. tetradactylum* was positively related to its body length. The larger fish predated on the larger prey as indicated by a positive relationship among standard length and mouth opening with the dimensions of all three major prey types (fish, shrimp and squid). Thus, *E. tetradactylum* of different sizes fed on different specific prey types according to its body size, resulting in less competition among fishes of different size classes in the same habitat. It is also obvious that size and mouth dimension of this species controlled the size of prey. This is also relevant for the 'optimum foraging theory' in which the cost/benefit ratio of catching prey is considered during foraging process by fish (*Gerking, 1994*; *Paul et al., 2017*).

*E. tetradactylum* and other co-existing species in Pattani Bay had moderate fullness index and vacuity index, indicating that most of their stomachs contained some food. These fish were considered specific feeders, based on their diet breadth and feeding on a narrow range of food type, except *J. belangerii*, *S. argus*, *S. lysan* and *S. commerson*. Trophic level analysis (TL) showed that most fish species had high trophic levels (TL > 3.50). *Froese & Pauly (2000)* suggested that the TL values of 2.0, 2.5 and 5.0 represented herbivores, omnivores and carnivores, respectively. These fish were thus categorized as species between omnivorous and carnivorous. The lowest trophic levels were *H. kelee* and *S. argus* which fed mainly on detritus and zooplankton. The estuarine fishes were mainly omnivorous and shared common resources with the flexibility to exploit temporary peaks of prey populations (*Ley, Montague & McIvor, 1994*).

Competition for food by fishes can lead to dietary switch, which may allow fish to adapt to its surrounding food (*Gerking, 1994*). Based on the results of diet overlap and cluster analysis, the eleven examined fish species can be divided into two trophic groups. *E. tetradactylum*, the target species, was grouped as fish/shrimp predator with some combination of zooplankton together with other species (*E. coioides, J. belangerii, S. commerson, S. lysan, O. ruber* and *L. russellii*). Other four co-existing species including *P. niger, E. jonesi, S. argus* and *H. kelee* avoided from feeding on fish and shrimp by shifting to smaller sized diets such as zooplankton, phytoplankton and detritus. This diet overlap and food partitioning between different species is important for community organization of fish in a particular habitat (*Krebs, 1989*) as well as for the survival of species (*Braga, Bornatowski & Vitule, 2012*; *Monteiro, Contente & Faria, 2018*).

## CONCLUSION

*E. tetradactylum* is an active and specific predator, with juvenile feeding largely on zooplankton especially *Acetes* and shrimp post larvae, small sized-fish predating mainly on penaeid shrimp, and medium and large-sized fishes feeding on a combination of penaeid shrimp, fish and squid. Different size, sex and habitat affected the fullness index and average number of food type predated by this fish species. Variation between diet composition and average number of food type between fish at transitional sex period with male and female fishes were recorded. The transitional sex fish predated almost entirely on other fishes (87.2%), whereas male and female fishes fed on penaeid shrimp and other fish, respectively. Fish size and mouth opening controlled the prey size consumed as the larger fish with larger mouth opening fed on the larger size of all prey types. In Pattani Bay, *E. tetradactylum* shared its diets with *E. coioides, J. belangerii, S. commerson, S. lysan, O. ruber* and *L. russellii* as teleost fish at high trophic levels (TL > 3.50), with penaeid shrimp being their main food The present study provided important information on the feeding habits of *E. tetradactylum* and its diet relationship with other co-existing fish species living in the same habitat of tropical region.

## ACKNOWLEDGEMENTS

Thanks are due to Faculty of Science and Technology, Prince of Songkla University for the use of all laboratories and equipment. Thanks are due to the crew of fisherman from the provinces of Pattani, Satun, Nakorn Si Thammarat and Samut Prakan who helped in fish sampling. Thanks are due to Wasina Rungruang, Arun Lohhem and Sofiyudin Maae for field sampling and laboratory work.

### Funding

This work was financially supported by TUYF Charitable Trust, Hong Kong and Ph.D. scholarship by Faculty of Science and Technology, Prince of Songkla University. The funders had no role in study design, data collection and analysis, decision to publish, or preparation of the manuscript.

### Grant Disclosures

The following grant information was disclosed by the authors:
TUYF Charitable Trust, Hong Kong and Ph.D. scholarship by Faculty of Science and Technology, Prince of Songkla University.

### Competing Interests

The authors declare there are no competing interests.

### Author Contributions

- Teuku Haris Iqbal conceived and designed the experiments, performed the experiments, analyzed the data, prepared figures and/or tables, authored or reviewed drafts of the article, and approved the final draft.

- Sukree Hajisamae conceived and designed the experiments, performed the experiments, analyzed the data, prepared figures and/or tables, authored or reviewed drafts of the article, and approved the final draft.
- Apiradee Lim analyzed the data, authored or reviewed drafts of the article, and approved the final draft.
- Sitthisak Jantarat performed the experiments, authored or reviewed drafts of the article, and approved the final draft.
- Wen-Xiong Wang conceived and designed the experiments, analyzed the data, prepared figures and/or tables, authored or reviewed drafts of the article, and approved the final draft.
- Karl W.K. Tsim conceived and designed the experiments, analyzed the data, authored or reviewed drafts of the article, and approved the final draft.

## Animal Ethics

The following information was supplied relating to ethical approvals (*i.e.*, approving body and any reference numbers):

institutional animal care and use committee, Prince of Songkla University.

## Data Availability

The raw data is available in the Supplemental File.

## Supplemental Information

Supplemental information for this article can be found online at http://dx.doi.org/10.7717/peerj.14688#supplemental-information.

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
