# Peer review of "Feeding habits of four-finger threadfin fish, Eleutheronema tetradactylum, and its diet interaction with co-existing fish species in the coastal waters of Thailand"

_PeerJ, doi:10.7717/peerj.14688_

## Round 0.1 · original submission · Major Revisions

Dear Authors

The reviewers have commented on your manuscript. You can find attached reports. Based on the comments and suggestions of the expert reviewers, a major revision is needed for your article.

While the reviewers found the article's topic valuable and noteworthy, they pointed out the seemingly simple but critical errors in the manuscript (e.g. mislabeling of tables, errors in statistical design, lack of details in methods, wrong citations, mistakes in used terms.)

I would like to request you check and correct the MS based on the reports and send them back.

Sincerely yours

·

Basic reporting

- L201: “Clarke & Gorley, 2006” is the reference of Primer v6.
- L328: “Mithun et al., 2017” is not in the reference list.

- L41: “Wootton, 1990” is provided as “Wootton, 1999”. The book is published the first in 1990, but there are no further editions (there have been only reissued version). Therefore, the reference should be provided as “Wootton, 1990”.
- L113: Please indicate the three sites for collection additional fish samples in Figure 1.
- Tables should be provided in order of the descriptions of the results.
- L205-210: Please refer to any Table for the descriptions.
- L211-212: This sentence should be referred to Table 6.

Experimental design

no comment

Validity of the findings

no comment

Additional comments

This manuscript describes feeding habitats of four-finger threadfin fish, and feeding relationships with co-occurring fish species. Overall, the manuscript is well structured, thus can be enough for publishing scientific paper. However, there also some issues before acceptance PeerJ. My specific comments indicate point-by-point in each of lines of the manuscript. Please consider my comment as well as those of other reviewer(s), and revised the manuscript according the comments.

L43-44: The word of “four-finger threadfin” is repeated in the sentence. Please consider to remove the latter word, i.e., “known as four-finger threadfins”.

L73: Generally, “piscivores” are common in the marine ecosystem, but not “cannibalism”. “Cannibalism” only occur in specific environmental conditions and/or for only a few species.

L190-193 as well as Figure 1 and 2: I am not sure how the authors can be fitting linear regression between area data and length data. The data of mouth opening area and prey dimension is two-dimensional, but length data is a single-dimensional. In this case, the data should be transformed before fitting regression (i.e., root transformation). Therefore, all relevant data and description should be changed using re-analyzed results accordingly.

L200: ANOVA and linear regression can not conduct. using PRIMER software package.

L226: The number of “131” is not same with those provided in Figure 1.

L260: “significant” → “high”??

L279: In Table 3, Acetes and juvenile shrimp are belonging to shrimp category.

Table 1: I just cannot believe how the authors can directly compare the prey composition between large items such as shrimp, fish and squid and phytoplankton. For example, volumetric contribution (purely two-dimensional according to scaled petri discs) of fish (distinctly three-dimensional) will be highly underestimated compared to phytoplankton (almost two-dimensional), thus the data of relative %IRI diet compositions cannot be acceptable. In this case, most researches have used weight data (i.e., dry weight), or cannot consider true prey item for phytoplankton, because most of fishes consume phytoplankton unintentionally in the process of ingesting other prey items.

Figure 6: According to the description in the text (L266-273), the feeding groups should be divided in three. In addition, to provide significant differences among the three feeding groups, further test is required such as SIMPROF test or ANOSIM pair-wise comparison.

Reviewer 2 ·

Basic reporting

Professional article structure, figures, tables. Raw data shared.
-> I found some mistakes in the name of tables (Tables 4, 5 and 6). The number of these tables do not match between the legend page and the legend above the table. Please check them carefully, and also correct the table names in the text if appropriate.

Experimental design

Methods described with sufficient detail & information to replicate.
-> Some details in the methods are not well explained. For example, the definition of the size class (e.g., juvenile, small-sized), sampling time (day or night), stage of digestion, are not fully explained. Authors should clarify these points.

Validity of the findings

Conclusions are well stated, linked to original research question & limited to supporting results.
-> I recommend authors to include the discussion of diet change in relation to sex change in the conclusion and abstract.

Additional comments

This manuscript describes the feeding habits of Eleutheronema tetradactylum with co-existing species in the coastal waters of Thailand. Authors have analyzed substantial numbers of specimens and analyzed these data by standard techniques. Generally, the manuscript is well structured, and I understand the importance of this study. However, I recommend minor revisions before the publication. Please check specific comments below. In particular, the issue of hermaphrodite (sex change) should be addressed more.

Abstract
L30: The genera of these species should be fully spelled out in the abstract.

Introduction
L80-82: Please clarify whether the full life history (from eggs/larvae to the spawning adults) is covered in the studied area.

Materials and methods
L89-110: Sampling time, an important factor for stomach fullness, should be addressed.

L118: Here, please provide other attributes of the specimens, especially for size and sex. Because the issue of hermaphrodite (sex change) is pointed out in Introduction (L55-58), it is important to describe the size range for each sex separately.

L130-L137: Dietary analysis. I understand that authors have used standard techniques, but the level of digestion is not referred to. In general, the %V is sensitive to the level of digestion among prey species. Please provide some information of digestion and its potential effects on results.

L133-135: Please elaborate more on “a modified point method and numerical method”.

L148-149: AF. I think it is better to define as “average number of food type”

L154: Please clarify that this “proportion” means %V or %N. (also at L182)

L186: Here, definitions of these potential factors should be fully described. In particular, definition of size class is very important. Please explain them with actual body size (e.g., juvenile fish: xx-xx cm, small-sized fish: xx-xx cm) with sex.

L192-193: Linear analysis. At least for the relationship between body size and mouth size (Fig. 2A), I recommended to use an allometry model (y = ax^b). It can better fit the relationship than the linear model.

Results

L206-208: English should be revised, for example: “The %IRI of Penaeid shrimp was 76.81%, 84.70%, and 70.02% in the stomachs from Samutprakarn, Satun, and Nakornsritammarat provinces, respectively.”.

L212: Tables 1 and 2 -> Tables 3 and 5? Number of tables should be carefully checked through the manuscript.

L214-216: ANOVA. I think there would be collinearity between size and sex due to the sex change during growth. Please address this point. In addition, the seasonal and spatial difference should be more discussed in the Discussion section (important).

L224: Table 6 -> Table 5?

L228-230: As noted above, allometry would better fit this relationship.

L231-242: Please address n and R^2 values when referring to the linear model in addition to P values.

L246: Table 1. Why “1” appears here? I think the tables should be numbered along with the appearance in the text.

L260: “five co-existing species” -> “six”? How about S. commerson?

Discussion

L285-289: I think this detailed diet menu should be described in Results.

L300: Where is the nursery habitat? Please clarify.

L323-324: This finding at the transient period should be one of the most important topics in this study in terms of life history of this species. So, I think this sentence should also be appeared in conclusion and abstract. Also, please expand discussion by comparing the present results with other species which exhibits sex change, if available.

Figure1: Please remove alphabets (A, B, and C) in the right panel if they are not relevant to the sampling design and analysis.

Table 1 and 3 (legend). “P. tetradactyum” -> “E. tetradactyum”?

Table 4: the number of table in the table page is “Table 5”. Please correct.

Table 5: the number of table in the table page is “Table 6”. Please correct.

Table 6: the number of table in the table page is “Table 4”. Please correct.

---

## Round 0.2 · Major Revisions

Dear Authors

I evaluated the revised version of your manuscript. I would like to thank you considered all the comments and suggestions of reviewers. Although there is no scientific deficiency in your article, I and the section editor decided this paper needed a round of strict English editing before acceptance.

Please have your manuscript checked by a proficient English speaker or language editing service.

Sincerely yours

·

Basic reporting

No comments

Experimental design

No comments

Validity of the findings

No comments

Additional comments

No comments

Reviewer 2 ·

Basic reporting

no comment

Experimental design

no comment

Validity of the findings

no comment

Additional comments

I went through the manuscript and acknowledged that authors have replied to all comments raised.
However, please reconsider regression analyses again. Authors have revised these analyses as allometry with square-root transformation for only Fig. 2. Maybe authors tried to respond to two reviewers’ comments (allometry by Rev. 2 (me) and, square-root by Rev. 1). But I think using either one is enough in this case.

---

## Round 0.3 · accepted · Accept

I evaluated the revised version of your manuscript. I would like to thank you considered the all comments and suggestions of reviewers. I am happy to say the current version of your manuscript is ready to publish.
Sincerely yours